# Optimizing Implementation of Preventive Chemotherapy against Soil-Transmitted Helminthiasis and Intestinal Schistosomiasis Using High-Resolution Data: Field-Based Experiences from Côte d’Ivoire

**DOI:** 10.3390/diseases10040066

**Published:** 2022-09-20

**Authors:** Jean T. Coulibaly, Eveline Hürlimann, Chandni Patel, Dieudonné K. Silué, Deles J. Avenié, Nadège A. Kouamé, Ulrich M. Silué, Jennifer Keiser

**Affiliations:** 1Swiss Tropical and Public Health Institute, CH-4123 Allschwil, Switzerland; 2University of Basel, CH-4003 Basel, Switzerland; 3Unité de Formation et de Recherche Biosciences, Université Félix Houphouët-Boigny, Abidjan 22 BP 770, Côte d’Ivoire; 4Centre Suisse de Recherches Scientifiques en Côte d’Ivoire, Abidjan 01 BP 1303, Côte d’Ivoire

**Keywords:** soil-transmitted helminthiasis, schistosomiasis, health district, sub-district, preventive chemotherapy

## Abstract

Background: Despite efforts to control neglected tropical diseases (NTDs) using preventive chemotherapy (PC), soil-transmitted helminthiases and schistosomiasis remain widely prevalent in sub-Saharan Africa. The current PC regimen in endemic settings is defined based on health district-level prevalence. This work aims to highlight the need for high-resolution data when elimination, rather than morbidity control, is the targeted goal. Methodology: Cross-sectional parasitological surveys were conducted from July to August 2019 and from September to October 2019, respectively, across the entire Dabou and Jacqueville health districts in southern Côte d’Ivoire. From every village, 60 school-aged children (6–15 years) were randomly selected and invited to provide one fresh stool sample, whereof duplicate Kato–Katz thick smears were prepared and read by two independent technicians. Principal Findings: 4338 school-aged children from 77 villages were screened from the Dabou (n = 2174; 50.12%, 39 villages) and Jacqueville (n = 2164; 49.88%, 38 villages) health districts. The prevalence of any soil-transmitted helminth (STH) infection was 12.47% and 11.09% in the Dabou and Jacqueville health districts, respectively. Species-specific district-level prevalence remained below 10%, varying between 0.51% (hookworm in Jacqueville) and 9.06% (*Trichuris trichiura* in Dabou). However, when considering sub-districts or villages only, several STH infection hotspots (five sub-districts with ≥20% and four villages with more than 50% infected) were observed. *Schistosoma mansoni* infection was found in less than 1% of the examined children in each health district. Conclusions/Significance: We conclude that keeping health district-level prevalence as a reference for PC implementation leaves many high-risk sub-districts or villages requiring PC (≥20% prevalence) untreated. To avoid maintaining those high-risk villages as STH reservoirs by skipping control interventions and jeopardizing the successes already achieved in STH control through PC during the past two decades, precision mapping is required. Further investigation is needed to assess cost-efficient approaches to implement small-scale disease surveillance.

## 1. Introduction

Soil-transmitted helminth (STH) infections are common infections in tropical and subtropical countries [1,2]. Four main species of STHs infect humans, namely *Ascaris lumbricoides* (roundworm), *Trichuris trichiura* (whipworm), *Ancylostoma duodenale* and *Necator americanus* (hookworms) [3,4]. An estimated 804 million people are infected with roundworm, 477 million with whipworm and 472 million with hookworms [5]. STHs are transmitted via ingestion of food and water contaminated with fecal material containing infective eggs or through skin penetration by hookworm larvae from walking barefoot on soil contaminated by human feces [3,4]. Consequently, infections mostly affect individuals living in communities with poor sanitation and hygiene, and inadequate access to safe and clean water [6]. Causing primarily subtle morbidity, STH infections rarely result in death. However, chronic infections may lead to digestive disorders and micronutrient deficiencies, including iron deficiency anemia, and thus, particularly have an impact on children’s physical growth and mental development, as well as on pregnancy outcomes [5,7,8].

Schistosomiasis is endemic in 54 countries, affecting approximately 240 million people worldwide, with up to 700 million people at risk of infection [9,10]. The disease is an intestinal or urogenital disease caused predominantly by infection with *Schistosoma mansoni*, *Schistosoma haematobium* or *Schistosoma japonicum*. Schistosomiasis is associated with anemia, chronic pain, diarrhea and undernutrition [11].

For both helminthiases, the main control strategy put forth by the World Health Organization (WHO) is preventive chemotherapy (PC) with albendazole or mebendazole and praziquantel, respectively, with the main goal to reduce infection intensity, and consequently, reduce STH- and schistosomiasis-related morbidity. Alongside PC, targeted health education and improved water and sanitation interventions are recommended [12,13]. PC is regularly implemented in many endemic countries, yet not all population groups at-risk are covered by the different programs. While for STH morbidity control, the range of targeted at-risk groups has been extended to preschool-age children (PSAC) and women of reproductive age (WRA, non-pregnant and pregnant after 1st trimester), school-age children (SAC; enrolled and non-enrolled) remain the main focus of schistosomiasis control, with the exception of adults with particular exposure (e.g., fishermen) [14,15]. 

In most sub-Saharan countries, the major targets and milestones of PC regularly evaluated by WHO are (i) to reach a minimal treatment coverage rate of 75% of targeted groups for morbidity control, as well as to shift the focus towards (ii) elimination as a public health problem [16]. The latter is defined as a proportion of less than 2% or 1% of moderate and heavy-intensity STH or heavy *Schistosoma* spp. infections, respectively, in the target population [17]. The frequency of PC (i.e., biannual or annual treatment rounds) depends, for both helminthiases, on the observed prevalence as determined at the district level. Consequently, endemic districts are classified into high-risk (≥50% prevalence), moderate-risk (STH: ≥20% up to <50%, schistosomiasis: ≥10% up to <50% prevalence) and low-risk (<20% STH and <10% schistosomiasis prevalence), respectively. Target populations in high-risk districts profit from biannual and annual mass drug administration (MDA) against STH and schistosomiasis, respectively. Moderate-risk districts receive annual STH treatment and praziquantel every second year for SAC. Lastly, STH low-risk districts are not considered for PC, while in districts with low schistosomiasis risk, SAC will be treated twice during their primary schooling age, and praziquantel has to be made available in dispensaries and clinics for the treatment of suspected cases [8,18]. These strategies may be cost-effective to achieve fast results for morbidity control, yet they remain inadequate to address transmission interruption or the elimination of the disease as a public health problem [19]. Considering prevalence at the district level does not take into account the focal geographical distribution of these diseases, and consequently, may ignore the reality of persistent hot spots [20]. Recently, WHO’s Expanded Special Project for Elimination of Neglected Tropical Diseases (ESPEN) has been put forth, calling for moving from the district to the community’s implementation of PC with the main goal of scaling up treatments to achieve 100% geographical coverage, leaving no one behind. Additional goals include building capacity to provide quality health services in order to make sure that those countries are able to interrupt transmission and reach their elimination goals [21].

This work aims to highlight the changes in treatment classification of implementation units when considering the sub-district instead of the district level, and the related implications for PC when elimination, rather than morbidity control, is the targeted goal.

## 2. Methods

### 2.1. Study Design

We designed cross-sectional eligibility surveys and conducted them from July to August 2019 and from September to October 2019, respectively, in the Dabou and Jacqueville health districts in southern Côte d’Ivoire. These cross-sectional surveys aimed to select suitable locations for the implementation of a randomized controlled trial (RCT) assessing the efficacy and safety of the co-administration of ivermectin and albendazole among individuals infected with *T. trichiura* [22]. 

### 2.2. Study Area, Population and Sampling Design

In Côte d’Ivoire, the health system is organized as a top-down pyramidal structure, as follows: the ministry of health at the top, followed by the health region, then, the health district, and finally, urban and rural health centers. A village and its related hamlets, or several villages together, share a rural health center managed by a nurse that sometimes also provides maternal health services by a midwife. In the study, we considered the rural health center (the last strata of the health pyramid) as the sub-district.

The Jacqueville and Dabou health districts are located 80 km and 50 km, respectively, from Abidjan, the economic capital of Côte d’Ivoire. Both health districts are characterized by a tropical climate with four seasons. A long rainy season (from April to July), during which, two-thirds of the annual rainfall occurs; a short dry season (from July to August); a short rainy season (from September to November) and a long dry season (from December to March). In both health districts, fishing, subsistence farming and cash crop production (e.g., palm oil, rubber and cacao) are the main economic activities. Many households lack access to permanent clean water, and open defecation is frequently practiced. 

For the purpose of our study, we aimed to cover the whole area of the two study districts rather than subsample a number of villages or schools, as is usually done in programmatic surveys to determine which MDA strategies to apply for the respective implementation units. In each district, all villages with less than 2500 inhabitants were targeted for the eligibility survey. We applied an adapted and previously used sampling method based on WHO recommendations for the collection of baseline information pertaining to helminth prevalence and intensity in the school-aged population within large-scale surveys [23]. From every village, 60 children aged between six and fifteen years old were randomly selected and invited to provide one fresh stool sample. A few small and close-by localities were pooled to achieve a subsample of 60 school-aged children. In view of about 80 localities in the two districts that correspond to the defined maximum population size, we estimated that we would screen up to 4800 children.

### 2.3. Ethical Consideration

Ethical approval for this work was obtained in 2015 (CNER, reference no. 037/MSLS/CNER-dkn). The purpose and procedures, including the potential risks and benefits for participants, were explained to the health authorities and in the native language of each village. Parents/guardians provided oral consent on behalf of their children, and oral assent was provided by all participants, as well. Participation was voluntary; hence, children could withdraw at any time without further obligations. At the end of the study, all soil-transmitted helminth- and *S. mansoni*-infected school-aged children were treated with a single oral dose of albendazole (400 mg) or praziquantel (40 mg/kg), respectively [24].

### 2.4. Field and Laboratory Procedures

The geographic coordinates of each village were registered using a handheld global positioning system (GPS) device (Garmin GPS map62 ST; Bucher+Walt SA, St-Blaise, Switzerland). In each village, a community health worker (CHW) was trained to conduct the census of 60 randomly selected children aged 6 to 15 years. The village was divided into four blocks. In each block, fifteen households were selected randomly, and in each household, one child of the targeted age group was selected. For each child, demographic characteristics were registered, namely age and sex. The primary source for obtaining a child’s age was their birth certificate. In cases where birth certificates were not available, parents were asked to provide a birth date. A unique identifier was attributed to each household and to the respective child.

The day before stool sample collection, children with oral parental consent were provided with an empty plastic container. They were asked to fill the container with a small portion of fresh morning stool. For both districts, stool samples from each village were transferred to the laboratory of the Methodist hospital of Dabou (HMD), subjected to duplicate Kato–Katz thick smears [25,26] and examined under a microscope by experienced laboratory technicians. Helminth eggs were enumerated and recorded for each species (i.e., *A. lumbricoides*, *T. trichiura*, and *S. mansoni*), separately. Hookworm eggs were recorded as such, since the common human-infecting species (i.e., *A. duodenale* and *N. americanus*) cannot be discriminated by light microscopy.

### 2.5. Data Analysis

Data were double entered into Microsoft Excel 2010 (Microsoft Corporation; Redmond, WA, USA) and cross-checked with EpiInfo version 3.5.4 (Centers for Disease Control and Prevention; Atlanta, GA, USA). Statistical analyses were done with STATA version 13.1 (Stata Corporation; College Station, TX, USA). Mean age and sex proportions were calculated per district. Helminth infection status was derived from species-specific egg counts, transformed into a binary variable (present/absent) and expressed as proportions for each parasite and for infection with any STH species. The prevalence was assessed at district, sub-district and village levels. Village and sub-district levels prevalence for *S. mansoni* and STHs were illustrated in maps using ArcGIS version 10.5.1 (ESRI, Redlands, CA, USA) and classified into risk categories according to WHO prevalence thresholds for PC (i.e., STH: high: ≥50%, moderate: 20–49.9%, low: <20%; *S. mansoni*: high: ≥50%, moderate: 10–49.9%, low: <10%) [8,18]. We compared the number of localities that would need MDA using the standard evaluation level of helminth prevalence (i.e., district level) with a classification based on sub-district and village level prevalence.

## 3. Results

### 3.1. Demographic Characteristics

Overall, 77 villages were included in this study, comprising 39 and 38 villages, respectively, in the Dabou and Jacqueville health districts. In total, 4338 school-aged children were screened, with one stool sample being analyzed using duplicate Kato–Katz thick smears, whereof 2174 were from Dabou (50.12%) and 2164 from Jacqueville (49.88%). The mean age of the participants was 9.19 years (standard deviation (SD) = ±2.36 years) and was comparable between districts with 9.10 years (SD = ±2.33 years) and 9.28 years (SD = ±2.38 years) in Dabou and Jacqueville, respectively. Inclusion of both sexes was well balanced, with 2061 (47.51%) females in total, comprising 996 (45.81%) females from Dabou and 1065 (49.21%) females from Jacqueville. In the Dabou health district, the 39 villages investigated belonged to 19 different sub-districts, whereas in the Jacqueville health district, the 38 villages included in the survey belonged to 14 sub-districts. 

### 3.2. Soil-Transmitted Helminthiasis

Table 1 shows the species-specific prevalence and any STH infection, according to the respective health district investigated. Prevalence of any STH infection was 12.47% [271/2174] and 11.09% [240/2164] in the Dabou and Jacqueville health districts, respectively. At each health district level, however, all species-specific prevalence was <10%, varying between 0.51% (hookworm in Jacqueville) and 9.06% (*T. trichiura* in Dabou). In the Dabou health district, we found three sub-districts with STH prevalence above 20%, namely Ahouya [38.05%, 113/297], Yassap [22.14%, 29/131] and Opoyounem [21.14%, 37/175] (Table 2). In these villages, *T. trichiura* was the predominant species. The remaining 16 sub-districts had an STH prevalence ranging between zero and 14.17%. In Jacqueville, one high [Tiagba: 94.23%, 49/52] and one moderate [Kouvé: 32.84%, 89/271] risk for STH infection sub-district were identified. All remaining 12 sub-districts showed STH prevalence below 20%. When analyzing at the village level, many hot-spots were observed in both health districts (Figure 1, Appendix A). In the Dabou health district, two villages located in the Ahouya sub-district had prevalence with any STH above 50%, namely Ahouya [75.44%, 43/57] and Akakro [70.00%, 42/60]. The majority of infections in these villages were infections with *T. trichiura*. In more detail, in Ahouya, 66.67% [38/57], 28.07% [16/57] and 1.75% [1/57] of individuals were infected with *T. trichiura*, *A. lumbricoides* and hookworm, respectively. Similar figures were observed in Akakro (prevalence of 65% [39/60], 21.67 [13/60] and 1.67% [1/60] with *T. trichiura*, *A. lumbricoides* and hookworm, respectively). Five and thirty-two villages in the same health district had STH prevalence classified as moderate and low risk categories, respectively. Similarly, in the Jacqueville health district, two villages, namely Tiagba [94.23%, 49/52] and Taboutou [70.69%, 41/58], were found to be high risk (>50%) STH localities. Three villages showed moderate STH risk (Téffrédji: 43.33% [26/60], Azahon: 42.86% [15/35] and Koko: 35.00% [21/60]), while 33 villages showed STH prevalence between zero and 19%. 

Figure 1 and Figure 2 show the high degree of heterogeneity and focality for STH infections, and highlight hotspots with prevalence above 50% at village level. The maps also identify a cluster of increased STH infection rates in the sub-district of Ahouya, Dabou health district (Figure 2A).

### 3.3. Intestinal Schistosomiasis

We found 0.74% [16/2174] and 0.46% [10/2164] school-aged children infected with *S. mansoni* in the Dabou and Jacqueville health districts, respectively. When considering sub-district or village levels, the prevalence of *S. mansoni* was never above 5% (see Appendix A). Infections were found in 17 out of 77 villages, with 10 situated in the Dabou and 7 in the Jacqueville health district.

## 4. Discussion

Over the past two decades, a significant reduction of the number of infected people with STHs and *Schistosoma* spp. has been achieved, largely thanks to MDA campaigns with donated anthelminthics [1,27,28]. However, these helminth infections continue to affect more than 1.5 billion people (~ 24% of the world’s population) among the poorest communities. STH infections and schistosomiasis are now targeted for elimination as a public health problem, defined as <2% proportion of STH infections of moderate and heavy intensity, and <1% proportion of schistosomiasis infections of heavy intensity, according to the most recent WHO roadmap for 2021–2030 [17]. To achieve this goal and the necessary transition, several obstacles and gaps should be considered and addressed. Truscott and colleagues have shown in their simulations how important persistent hot spots for STH infections can be for MDA effectiveness, and thus, they call for more accurate assessment of small-scale heterogeneity [29]. For schistosomiasis, the WHO’s ESPEN program has already analyzed MDA treatment gaps in 2019–2020 and provided sub-district datasets for at least 35 African countries, clearly advocating for sub-district implementation of PC [30].

The current work aims to highlight the changes in prevalence classification when analyzing at the sub-district instead of the health district level, and the implications for PC interventions. 39 and 38 villages, respectively, were involved in the current study in the Dabou and Jacqueville health districts, with more than 4300 children screened for appraisal of infection rates with STHs and intestinal schistosomiasis. We found a very low prevalence (<1%) of intestinal schistosomiasis in both health districts. The profile did not change when considering the prevalence at sub-district or village levels. If STH prevalence was considered at the district level only, both health districts did not require PC intervention (<20%). However, considering data at the sub-district level only, five sub-districts (covering eighteen villages) would need PC interventions for morbidity reduction that are not subjected to control activities according to the risk classification currently in place. At an even lower level, four villages would even require bi-annual rounds of STH treatment (≥50% prevalence). The produced maps (Figure 2 and Figure 3) further highlight the high degree of focality with low- and high-risk areas lying close to one another. 

### 4.1. Intestinal Schistosomiasis Is Hypoendemic in Investigated Health Districts 

Based on the current schistosomiasis prevalence, the health districts investigated do not require PC. Further investigations are required to deepen our understanding of the schistosomiasis profile in the study settings, as villages with the same ecological profile have shown to be at risk of *S. mansoni*. Bayesian risk mapping and model-based estimation of historical and recent epidemiological schistosomiasis data from Côte d’Ivoire allowed for the identification of high-risk settings for both intestinal and urogenital schistosomiasis at the country level [31]. The main foci for *S. mansoni* infection were estimated to be found in the health district of Agboville (southern part) [32] in areas in the southwestern part of the country [33], as well as in areas in the close north of the economic capital, Abidjan [31]. According to modelling data from Chammartin et al. [31], our study settings have not been determined as high-risk settings for intestinal schistosomiasis, which is in line with our observations. We did not find any previous data on schistosomiasis prevalence in either health district to incriminate the impact of PC. It is likely that environmental factors rendered our study district less suitable for schistosomiasis transmission, as highlighted by Walz et al., who found significant relations between a habitat suitability index and infection prevalence in study sites of Côte d’Ivoire [34]. Our diagnostic approach used to identify *S. mansoni* infections—duplicate Kato–Katz thick smear readings—is known to be less sensitive, particularly in low endemicity settings, where infection intensities are expected to be low. It is, therefore, likely that more infections would have been detected if point-of-care circulating cathodic antigen (POC-CCA) cassette tests or PCR-based molecular diagnostics would have been applied [35]. However, to date, control programs still rely on parasitological stool examination techniques, as used in this study. The introduction of more sensitive diagnostic tools implies adaption of prevalence cut-offs for control strategy implementation. In the case of POC-CCA, the WHO recommends 30% prevalence by this test as equivalent to a 10% Kato–Katz-based prevalence [36]. Consequently, we still consider Dabou and Jacqueville as hypoendemic for *S. mansoni* infections. 

### 4.2. Spatial Heterogeneity of Soil-Transmitted Helminthiasis

In both investigated health districts, we observed a low overall STH prevalence (<20%), most of them probably resulting from the efforts by the national control programme’s annual MDA, using albendazole against STHs or the co-administration of albendazole and ivermectin against lymphatic filariasis and onchocerciasis to high-risk district populations across the country during the past decade [37,38]. Both health districts studied would not require PC if the current treatment recommendations were applied. However, our comprehensive cross-sectional screening allowed for the detection of several hotspots and highlighted the high degree of small-scale heterogeneity in the distribution of STHs. Further investigations are warranted to characterize and explain these hotspots, including investigation on treatment adherence [39,40], sanitation [41], environmental suitability, behavioral aspects [42,43] and drug susceptibility. Of note, only 10 years ago, hookworm was the predominant STH species in Côte d’Ivoire [23]. *T. trichiura* has since become predominant in our study areas, which may also be a result of the current drug regimens used during PC (i.e., single dose of albendazole and mebendazole), which show limited efficacy against this species [44]. Additionally, the identified hotspots need special attention for factors hindering effective control, such as impaired drinking water source management (e.g., Ahouya cluster) and geographic obstacles for sanitation improvement (e.g., Tiagba with high ground water levels and direct sewage drainage into the surrounding waterbodies).

### 4.3. Need for Sub-District Implementation of PC and How to Get There

Our cross-sectional large-scale study highlights that many at-risk villages are overlooked by the current treatment recommendation reference, which is set at the district level, but that considering sub-district level prevalence for STHs can provide a more accurate picture of the transmission situation [45]. In our case, we found that both health districts, based on their respective STH prevalence, did not require PC implementation, yet 12 villages with moderate to high risk (>20% prevalence) were identified. Therefore, moving from morbidity control to the elimination of STHs as a public concern will require precision mapping in order to highlight the hotspots [43]. The mapping of STH and schistosomiasis at a smaller scale requires further financial, logistic and human resources. On the other hand, precision mapping might be a cost-effective tool, as it will prevent morbidity and result in rational utilization of the donated tablets for PC. However, solutions to optimize disease surveillance in a more cost-effective manner are needed. One way forward to save resources with regard to diagnostic efforts could be with sample-pooling strategies [46,47]. Large-scale implementation of this approach is still warranted, including cost-effectiveness assessment. Another option to generate sub-district data and maps would be to foster the reuse of existing data. WHO’s ESPEN program (https://espen.afro.who.int/ accessed on 12 June 2021) encourages control program managers to use any available data from robust population-based research studies, health centers and partners (including NGOs), as well as environmental data to better address data gaps. A concerted approach between partners from academia, private and governmental structures involved in schistosomiasis and STH research or control, but also all related fields impacting these diseases, such as human development, water, sanitation and hygiene, and ecology, could help to reach this objective. To support such a data sharing initiative, control program managers could generate an electronic platform and ensure its standardization among researchers. Primary data of interest would be prevalence data, but also information on environmental and sanitation conditions could highlight potential exposure for the disease. 

We conclude that schistosomiasis is currently not a public health problem in the Dabou and Jacqueville districts in Côte d’Ivoire. When considering the prevalence of STH infection at the health district level as the implementation unit for PC, neither health district requires any PC intervention. However, at the sub-district level, many villages are in need of PC implementation. Our findings corroborate the call for a transition from the district to the sub-district level as an implementation unit for PC targeting STH infections, particularly where elimination is the goal. The identification of cost-effective solutions to generate the underlying small-scale data for its implementation are of high priority. 

## Figures and Tables

**Figure 1 diseases-10-00066-f001:**
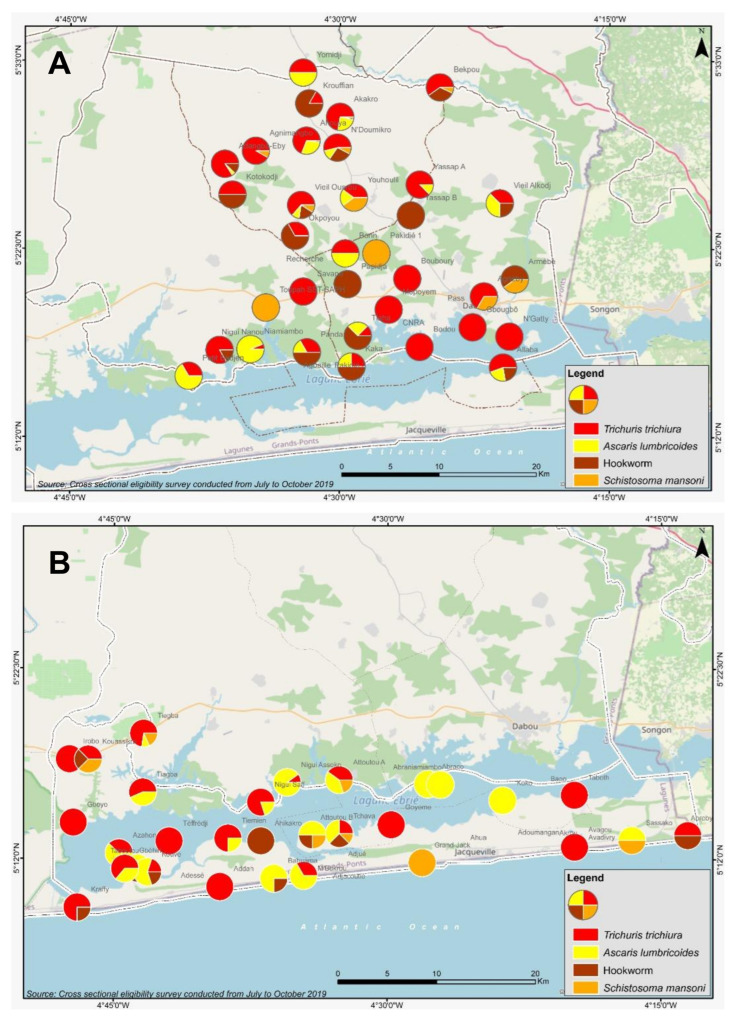
Species composition of STH (*A. lumbricoides, T. trichiura* and hookworm) and *S. mansoni* infections at village level in the Dabou (**A**) and Jacqueville (**B**) health districts. Pie charts represent all identified helminth infections (only positive cases) in the respective villages where any of the investigated infections were found, while its segments provide the proportion of each species. Villages without geographical coordinates and without any helminth infection detected are not depicted (Dabou: n = 8, Jacqueville: n = 9).

**Figure 2 diseases-10-00066-f002:**
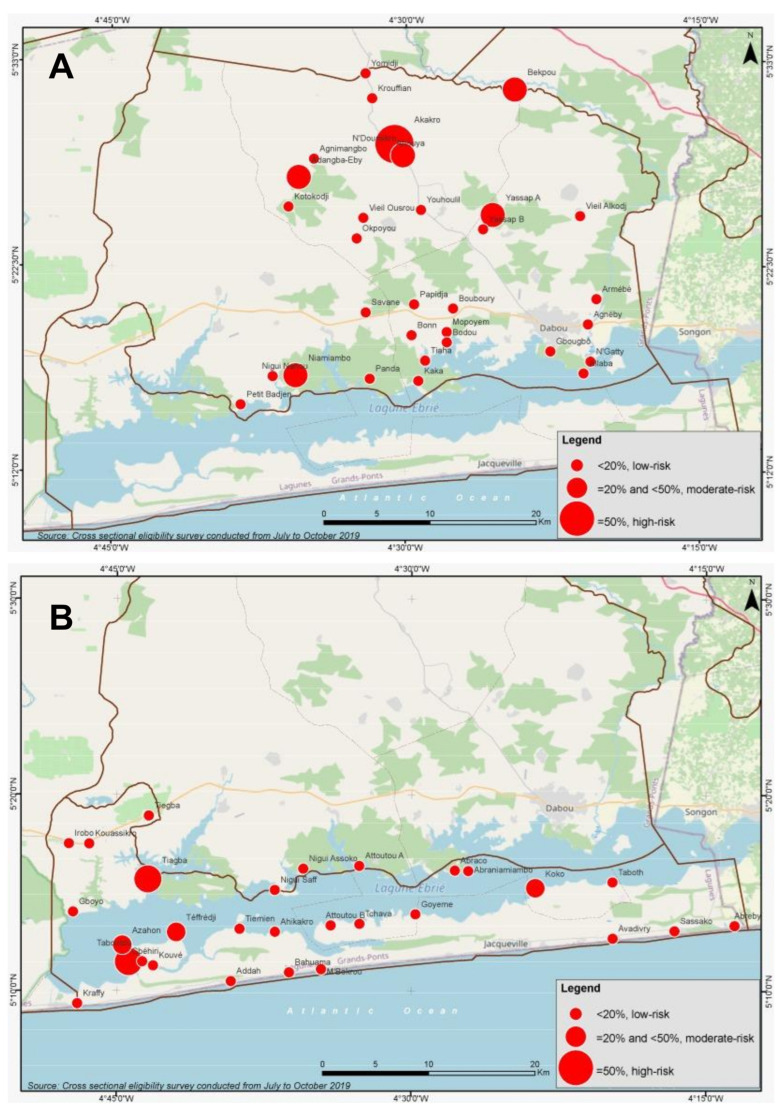
Geographic distribution and prevalence of infection with any STH species at village level in Dabou (**A**, n = 31 villages) and Jacqueville (**B**, n = 29 villages) health districts. Current treatment classifi-cation for district level prevalence according to WHO [8] considering school-age children (SAC) that remain the major target group for mass drug administrations: (i) <20%, low-risk, no need of preventive chemotherapy; (ii) ≥20% and <50%, moderate-risk, treatment of all SACs (enrolled and not enrolled) once each year; (iii) ≥50%, high-risk, treatment of all SACs (enrolled and not enrolled) twice each year.

**Figure 3 diseases-10-00066-f003:**
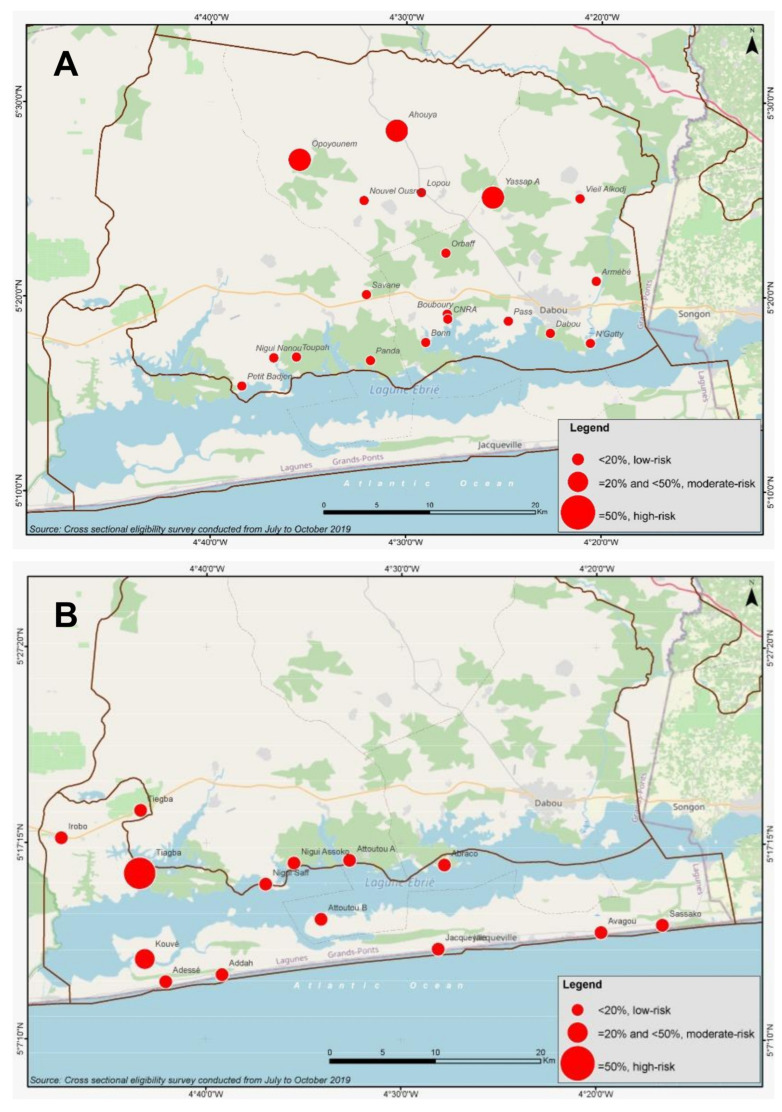
Geographic distribution and prevalence of infection with any STH species at sub-district level in the Dabou (**A**) and Jacqueville (**B**) health districts. Prevalence is classified according WHO thresholds set for preventive chemotherapy (PC) intervention [8]: (i) <20% low-risk, no need of PC; (ii) ≥20% and <50%, moderate-risk, treatment of all school-age children (enrolled and not enrolled) once each year; (iii) ≥50%, high-risk, treatment of all school-age children (enrolled and not enrolled) twice each year.

**Table 1 diseases-10-00066-t001:** Prevalence of *S. mansoni*, soil-transmitted helminth species and infection with any kind of STH (single- or multiple) infection in school-aged children from Dabou and Jacqueville health districts.

Parasite Species	Dabou (N = 2174)		Jacqueville (N = 2164)	
	No. of Infected Children (%)	95% CI	No. of Infected Children (%)	95% CI
**Intestinal Schistosomiasis**				
*S. mansoni*	16 (0.74)	0.38–1.10	10 (0.46)	0.18–0.75
**Soil-Transmitted Helminths (STHs)**				
*Trichuris trichiura*	197 (9.06)	7.85–10.27	160 (7.39)	6.29–8.50
*Ascaris lumbricoides*	68 (3.13)	2.40–3.86	130 (6.01)	5.01–7.01
Hookworm	50 (2.30)	1.67–2.93	11 (0.51)	0.21–0.81
Any STH	271 (12.47)	11.08–13.86	240 (11.09)	9.77–12.41

**Table 2 diseases-10-00066-t002:** STH and *S. mansoni* infection prevalence by sub-district and district.

		*S. mansoni*		*Trichuris trichiura*		*Ascaris lumbricoides*		Hookworm		Any STH	
Sub-District	No. Examined	No. Infected	%	No. Infected	%	No. Infected	%	No. Infected	%	No. Infected	%
** *Dabou* **											
Ahouya	297	3	1.01	94	31.65	34	11.45	14	4.71	113	38.05
Yassap	131	1	0.76	23	17.56	2	1.53	9	6.87	29	22.14
Opoyounem	175	1	0.57	34	19.43	1	0.57	6	3.43	37	21.14
Toupah	120	3	2.50	1	0.83	16	13.33	0	0.00	17	14.17
Nouvel Ousrou	60	1	1.67	7	11.67	1	1.67	2	3.33	8	13.33
Vieil Alkodj	60	0	0.00	3	5.00	3	5.00	2	3.33	8	13.33
N’Gatty	111	0	0.00	8	7.21	2	1.80	2	1.80	11	9.91
Bonn	172	0	0.00	4	2.33	5	2.91	7	4.07	16	9.30
Nigui Nanou	57	0	0.00	5	8.77	0	0.00	1	1.75	5	8.77
Lopou	52	2	3.85	2	3.85	1	1.92	0	0.00	3	5.77
Dabou	118	1	0.85	6	5.08	0	0.00	0	0.00	6	5.08
Armébé	60	2	3.33	0	0.00	0	0.00	3	5.00	3	5.00
Petit Badjen	60	0	0.00	1	1.67	2	3.33	0	0.00	3	5.00
Bouboury	180	0	0.00	6	3.33	0	0.00	0	0.00	6	3.33
Panda	120	0	0.00	2	1.67	1	0.83	3	2.50	4	3.33
Savane	176	0	0.00	1	0.57	0	0.00	1	0.57	2	1.14
CNRA	60	0	0.00	0	0.00	0	0.00	0	0.00	0	0.00
Orbaff	105	2	1.90	0	0.00	0	0.00	0	0.00	0	0.00
Pass	60	0	0.00	0	0.00	0	0.00	0	0.00	0	0.00
Total Dabou	2174	16	0.74	197	9.06	68	3.13	50	2.30	271	12.47
** *Jacqueville* **											
Tiagba	52	0	0.00	47	90.38	36	69.23	0	0.00	49	94.23
Kouvé	271	0	0.00	72	26.57	38	14.02	2	0.74	89	32.84
Nigui Assoko	60	0	0.00	1	1.67	9	15.00	0	0.00	9	15.00
Tiegba	60	2	3.33	8	13.33	1	1.67	0	0.00	9	15.00
Abraco	120	0	0.00	2	1.67	10	8.33	0	0.00	12	10.00
Nigui Saff	59	0	0.00	4	6.78	1	1.69	0	0.00	5	8.47
Attoutou B	154	2	1.30	3	1.95	5	3.25	3	1.95	11	7.14
Jacqueville	357	1	0.28	0	0.00	21	5.88	0	0.00	21	5.88
Addah	295	0	0.00	6	2.03	6	2.03	2	0.68	14	4.75
Adessé	170	0	0.00	7	4.12	0	0.00	1	0.59	8	4.71
Attoutou A	53	1	1.89	2	3.77	2	3.77	0	0.00	2	3.77
Irobo	180	3	1.67	5	2.78	0	0.00	2	1.11	6	3.33
Sassako	120	1	0.83	1	0.83	1	0.83	1	0.83	3	2.50
Avagou	213	0	0.00	2	0.94	0	0.00	0	0.00	2	0.94
Total Jacqueville	2164	10	0.46	160	7.39	130	6.01	11	0.51	240	11.09

## Data Availability

De-identified individual participant data reported in this research article are available upon request from the corresponding author after all findings are published. Data will be shared after the approval of a proposal by the authors for legitimate scientific purposes.

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
