# Peer review of "Optimizing Implementation of Preventive Chemotherapy against Soil-Transmitted Helminthiasis and Intestinal Schistosomiasis Using High-Resolution Data: Field-Based Experiences from Côte d’Ivoire"

_diseases, 2022, doi:10.3390/diseases10040066_

Round 1

Reviewer 1 Report

The authors present a study of the field of early prevention and control efforts for neglected tropical diseases (NTDs) using preventive chemotherapy (PC). Current PC in endemic areas are defined based on health district prevalence levels. The authors aim to highlight the need for high-resolution data when elimination rather than control of disease is the goal.
A cross-sectional parasitological survey methodology was used; these were conducted between July and August 2019 and September and October 2019 in the localities of
The entire health districts of Dabou and Jacqueville in southern Côte d'Ivoire.
From each village, 60 school-aged children (6-15 years) were randomly selected and asked to provide one fresh stool sample, from which a duplicate was taken. Kato-Katz thick swabs were prepared and read using two independent techniques.

A total of 4338 school-aged children from 77 villages were examined. From Dabou (n = 2174; 50.12%, 39 villages) and Jacqueville (n = 2164; 49.88%, 38 villages). The prevalence of any STH infection was 12.47% and 11.09% in Dabou and Jacqueville health districts, respectively,
respectively.
The authors concluded that maintaining health district-level prevalence as a benchmark for PC implementation leaves many high-risk areas at the subdistrict or village level requiring PC (≥20% prevalence) untreated. It appears that further investigation is needed to assess cost-effective approaches to implementing small-scale disease prevention surveillance.

This paper deals with the statistical evaluation of the treated samples, using two independent methods, and evaluates the factors of interest on a sufficient sample of the study population. The authors declare the optimization of the implementation of preventive chemotherapy . This section would deserve a methodological description and description of the new approach in the methodology and its advantages over the current solution. This part is not highlighted in the text.
I recommend to pay attention when completing the study to the method and methodologies of optimization.

Reviewer 2 Report

The authors of the article: “Optimizing implementation of preventive chemotherapy against soil-transmitted helminthiasis and schistosomiasis using high-resolution data: Field-based experiences from Côte d’Ivoire” are sharing field-based research experiences from West Africa. They show that the species-specific prevalence of Soil-Transmitted Helminthiasis at the district level is low but high enough at the sub-districts and villages level for Preventive Chemotherapy intervention according to the World Health Organization guidelines. However, the study found that the prevalence of intestinal schistosomiasis is low in the study areas. Researchers concluded that precision mapping is required to keep infection hotspots on PC intervention. The study is addressing an important scientific question and the data presented here are unique.  Therefore, this article presents some novelty in its design and the data gathered and adds specific information to the ultimate goal of the elimination of helminthiasis and schistosomiasis. The study is important for decision-makers, epidemiologists, parasitologists, and healthcare providers in the study areas as well as in other countries in Africa. However, there are some methodological aspects of the design that are missing and some data need some more clarifications/explanations so that the readers can benefit more from the manuscript.

The authors should address the following:

1.    The abbreviation STH, used for the first time in the abstract needs its full form. Some readers will read the abstract before reading the entire paper. It is always a good idea to allow readers to get a better idea of the rest of the paper. See Line 27.

2.    I would suggest adding “intestinal” before schistosomiasis in the title since the authors are only addressing S. mansoni.

3.    Line 124. The authors used the word “open defection” and probably wanted to write “open defecation”.  

4.    How was the sample size computed or estimated? Some studies use the statistical n formula where n is the sample size, Z is the standard score of the level of confidence, P is the expected prevalence, and d is the margin of error. Why did the authors choose to screen 4,338 SAC?

5.    Line 191. What does XXX mean?

6.    Table 1. I would replace hookworm with the name of the species studied here to keep things consistent.

7.    Table 1. I did not understand why any STH is 271 and 240 in Dabou and Jacqueville, respectively. Can the authors help me understand these numbers?

8.    Table 2. Why hookworm? Why not use the name of the species as indicated for the other species?

9.    Based on the technique used to diagnose intestinal schistosomiasis, the authors believe that this disease is hypoendemic in investigated health districts. I understand that KK is the standard field method of detecting Schistosoma mansoni in patient stool samples by microscopy. However, KK is known to miss some infections, especially those of light intensity according to previous studies (Berhe et al., 2004; Clemens et al., 2018). Alternatively, the point-of-care circulating cathodic antigen (CCA) is more sensitive than KK, especially when the intensity is low. Whether the test is performed in the laboratory or in the field, CCA outperformed KK in a previous study in Burundi (Clements et al., 2018; Fuss et al., 2020). Also, real-time PCR has high diagnostic accuracy. In the light of these studies, what are the comments of the authors?

10. The study is entirely missing any statistical significance. Is statistical analysis irrelevant to this design? Please, can you explain?

Reviewer 3 Report

The article is well written and I have some minor grammatical comment please see attached PDF for that.

Round 2

Reviewer 1 Report

-